# Measurements of Electrodermal Activity, Tissue Oxygen Saturation, and Visual Analog Scale for Different Cuff Pressures

**DOI:** 10.3390/s24030917

**Published:** 2024-01-31

**Authors:** Youngho Kim, Incheol Han, Jeyong Jung, Sumin Yang, Seunghee Lee, Bummo Koo, Soonjae Ahn, Yejin Nam, Sung-Hyuk Song

**Affiliations:** 1Department of Biomedical Engineering, Yonsei University, Wonju 26493, Republic of Korea; han0506000@naver.com (I.H.); tachanka7758@yonsei.ac.kr (J.J.); abbey0909@yonsei.ac.kr (S.Y.); fhrm502@yonsei.ac.kr (S.L.); beommo@yonsei.ac.kr (B.K.); 2Institute of Smart Rehabilitation Engineering and Assistive Technology, Dong-Eui University, Busan 47340, Republic of Korea; asj@deu.ac.kr; 3Department of Clinical Development, Angel Robotics, Seoul 04798, Republic of Korea; yejin.nam@angel-robotics.com; 4Department of Robotics & Mechatronics, Korea Institute of Machinery & Materials, Daejeon 34103, Republic of Korea; shsong@kimm.re.kr

**Keywords:** comfort, visual analog scale (VAS), electrodermal activity (EDA), tissue oxygen saturation (StO_2_), binding part, cuff pressure

## Abstract

The quantification of comfort in binding parts, essential human–machine interfaces (HMI) for the functioning of rehabilitation robots, is necessary to reduce physical strain on the user despite great achievements in their structure and control. This study aims to investigate the physiological impacts of binding parts by measuring electrodermal activity (EDA) and tissue oxygen saturation (StO_2_). In Experiment 1, EDA was measured from 13 healthy subjects under three different pressure conditions (10, 20, and 30 kPa) for 1 min using a pneumatic cuff on the right thigh. In Experiment 2, EDA and StO_2_ were measured from 10 healthy subjects for 5 min. To analyze the correlation between EDA parameters and the decrease in StO_2_, a survey using the visual analog scale (VAS) was conducted to assess the level of discomfort at each pressure. The EDA signal was decomposed into phasic and tonic components, and the EDA parameters were extracted from these two components. RM ANOVA and a post hoc paired *t*-test were used to determine significant differences in parameters as the pressure increased. The results showed that EDA parameters and the decrease in StO_2_ significantly increased with the pressure increase. Among the extracted parameters, the decrease in StO_2_ and the mean SCL proved to be effective indicators. Such analysis outcomes would be highly beneficial for studies focusing on the comfort assessment of the binding parts of rehabilitation robots.

## 1. Introduction

Rehabilitation robots, or exoskeleton robots, detect user intentions and provide assistance by attaching robotic limbs to humans. Along with the development of robotics technology, there has been active research into the design, structure, and control of rehabilitation robots [1]. Research on rehabilitation robots is advancing rapidly in the rehabilitation, medical, and power augmentation fields [2,3]. Concurrently, the market demand for rehabilitation robots is growing [3]. However, as this field develops, issues concerning the comfort of human–machine interfaces are becoming increasingly significant [2]. At present, there is a lack of objective evaluation criteria and guidelines for assessing the comfort of rehabilitation robots. In rehabilitation robotics, applying force to control the robot and maintain its attachment to the user is inevitable. However, excessive force can cause discomfort, pain, and injury from the device, and in severe cases, it can lead to skin problems such as pressure ulcers [4]. Consequently, assessing the fit of rehabilitation robots is crucial to ensure user comfort and prevent injuries caused by excessive pressure.

The VAS has been widely used for evaluating pain or discomfort [4,5]. Daly et al. [4] used a VAS as a pain score to determine a correlation with the maximum pressure of sockets of upper-limb prostheses. In their study, the VAS was normalized by the subject’s average value to minimize the threshold bias of individuals’ perception of discomfort. Meyer et al. [5] created questionnaires that included the a VAS, the Likert scale, and a numerical rating scale to assess the usability of wearable robots and reported that the VAS, Likert scale, and open text were mainly preferred by subjects.

Several studies quantified discomfort based on multiple bio-signals, as the autonomic nervous system (ANS) is the main body in response to stimuli. However, only a few studies used EDA signals and StO_2_, such as Léger et al. [6]. In their study, they evaluated workstations, but there are only a handful of studies that applied these signals to other types of binding parts in rehabilitation robots.

Many studies show that EDA signals are the most reliable for measuring discomfort [7,8,9,10]. EDA, which is also known as GSR (galvanic skin response), refers to the electrical phenomenon caused by sweat secretion from the skin and appendages [11]. It is measured on the skin surface, primarily on the palms, fingers, and soles, where sweat glands are distributed [12,13]. When the body is subjected to physical stimuli as well as various mental stimuli, such as tension and excitement, sweat secretion from sweat glands increases in response to the somatosensory and sympathetic nervous systems (SNS) [13], which reduces skin resistance and increases the strength of the EDA signal. The EDA signal can be decomposed into phasic and tonic components. The phasic component is a rapid response to external stimuli, while the tonic component represents a slowly changing baseline of the signal [13]. Several studies were conducted based on these characteristics of EDA. Kong et al. [14] applied electrical stimulation to the right forearm, decomposed the EDA signal using a high-pass filter with a cutoff frequency of 0.05 Hz, and developed an algorithm to detect the phasic component and determine the pain based on stimulation intensity. Kim et al. [15] measured EDA on the left palm by applying pressure stimulus to the left scapula, observing a linear increase in the maximum SCR (skin conductance response) amplitude as the pressure intensity increased. Posada-Quintero et al. [16] found that the mean SCL (skin conductance level) and the number of SCR peaks significantly increased when applying physical stimuli, postural stimulation, and the cold pressor test. Hosseini et al. [17] demonstrated that 87 features extracted from EDA signals in the WESAD (wearable stress and affect detection) dataset were sufficient to classify stress and non-stress groups with over 80% accuracy. They also found that five features—the mean SCL, maximum SCL, number of SCR peaks, maximum SCR amplitude, and standard deviation of SCR rise time—were adequate to classify the two groups with 97% accuracy.

Numerous studies explored the effects of tissue compression on blood circulation and microvasculature [18,19,20]. Linnenberg et al. [21] reported that high-pressure tissue compression during the use of rehabilitation robots can cause discomfort, pain, or soft tissue damage owing to ischemia. Their findings revealed a decrease in tissue oxygenation of 4.1% per minute at the robot’s binding parts during rest. In some research focusing on wearable robotic bindings for the lower extremities, it was observed that the thigh caused more pain and exhibited a significant increase in oxygen saturation at the same pressure compared with the shank, particularly during standing or walking activities [22,23,24]. In addition, Nam et al. [24] conducted a study to measure StO_2_ and quantified the comfort of a standing supportive rehabilitation robot at different binding sites. This study employed near-infrared spectroscopy (NIRS), which utilizes near-infrared light at wavelengths of 630–1300 nm. NIRS penetrates tissues to depths of 1–3 cm and measures changes in the concentration of oxygenated and deoxygenated hemoglobin; therefore, it is effective for the continuous and non-invasive monitoring of the relevant area.

In this foundational research, we employed a pneumatic cuff for pressure stimulation—a commonly used binding component in rehabilitation robots—to analyze the effects of binding forces through bio-signals such as EDA and StO_2_. This study aims to elucidate the relationship between pressure and bio-signals in rehabilitation robots, potentially making a significant contribution to the design and user interface of such devices.

## 2. Materials and Methods

### 2.1. Participants

This study included 23 healthy adults (21 males and 2 females, 24.1 ± 1.0 years old, 173.6 ± 6.1 cm, 72.3 ± 12.4 kg) with no neurological or musculoskeletal abnormalities in the lower extremities. Measurements were conducted on participants in sensor attachment sites that were free of skin pigmentation and on those with similar skin tones to ensure smooth NIRS signal measurement [25,26]. Participants were instructed to refrain from eating for 1 h prior to the test and to limit caffeine, nicotine, and alcohol for 5 h prior to the test. All participants fully understood the experiment’s procedures and signed an informed consent form. The experimental procedures were approved by the Institutional Review Board of Yonsei University (1041849-202304-BM-070-02).

### 2.2. Questionnaire

To assess discomfort, participants were asked to complete a questionnaire during the application of pressure. A visual analog scale (VAS) was included in the questionnaire to indicate pain at each pressure from 0 (no pain) to 10 (the worst possible pain), and for normalization, 60 kPa was applied to the right thigh for 5 s before the experiment to establish a baseline [27].
(1)$$Nomalized\;VAS=\frac{VAS\;of\;pressure\;condition}{VAS\;at\;60\;kPa}\times 100\;[\%]$$

### 2.3. Equipment

#### 2.3.1. Pneumatic Cuffs and Pressure-Applying Devices

Pneumatic cuffs (no-pinch single cuff, 86 cm × 10 cm (DTC-S07), DS Maref, Gunpo, Republic of Korea) were used. The cuffs were worn on both legs, and only the cuff worn on the right leg was subjected to pressure stimulation using a pressure applicator (MoorVMS-PRES; Moor Instruments, Axminster, UK).

#### 2.3.2. EDA Sensors

An EDA sensor (GSR100C, BIOPAC, Goleta, CA, USA), used in conjunction with the MP150 (BIOPAC, Goleta, CA, USA), measured the change in skin conductance due to a pressure stimulus applied to the adhesions. The EDA signal was measured at a sampling rate of 1 kHz and then separated into phasic and tonic components by taking the HPF (f_c_ = 0.05 Hz). The following characteristic values were extracted and analyzed from each component [28].

Mean SCL: Mean value of SCL during pressure stimulation;Max amplitude: Maximum value of SCR amplitude during pressure stimulation;SCR counts: Number of SCR peaks during pressure stimulation.

#### 2.3.3. Near Infrared Spectroscopy (NIRS) Sensor

A NIRS sensor (MoorVMS-NIRS, Moor Instruments, Axminster, UK) was used to monitor changes in blood flow underneath the cuff. Measurements were taken at a sampling rate of 5 Hz using near-infrared light at 750 nm and 850 nm. NIRS can measure oxyHb and deoxyHb concentrations. Tissue oxygen saturation (StO_2_) was determined using Equation (2).
(2)$${StO}_{2}=\frac{oxyHb}{oxyHb+deoxyHb}\times 100\;[\%]$$

The decrease in StO_2_ was analyzed using an average interval of 30 s from the time of pressure application.

### 2.4. Experimental Protocol

#### 2.4.1. Experiment 1: Measurement of EDA for Different Pressures

In Experiment 1, 13 adults (11 males, 2 females, 24.8 ± 0.7 years old, 174.5 ± 6.0 cm, 72.9 ± 14.2 kg) were tested. Pneumatic cuffs were worn on both thighs to provide pressure stimulation, and Ag/AgCl electrodes were attached to the thumb and cubital fossa of the left palm to measure EDA signals (Figure 1a) [13]. The experiment was conducted in a sedentary state. To prevent the subject’s adaptation to the experiment, three different pressures of 10, 20, and 30 kPa were randomly applied to the pneumatic cuff on the right thigh for 1 min each. A VAS was obtained after each pressure was removed, and a 5 min rest period was given to the subjects.

#### 2.4.2. Experiment 2: Measurement of StO_2_ and EDA for Different Pressures

In Experiment 2, 10 adult males (23.2 ± 1.0 years old, 173.6 ± 6.0 cm, 72.3 ± 12.4 kg) were recruited. NIRS sensors were attached to the right great trochanter, and 2/3 of the lateral epicondyle (Figure 2a), and EDA sensors were attached to the same locations as in Experiment 1 [13]. Pneumatic cuffs were worn on both thighs, and three different pressures of 10, 20, and 30 kPa were applied to the right thigh for 5 min. Similarly, a VAS was obtained after each pressure was removed, and a 5 min rest period was given to the subjects.

### 2.5. Statistical Analysis

Repeated measures ANOVA (RM ANOVA) and a post hoc paired *t*-test were used to determine significant differences as the stimulation pressure increased [29]. RM ANOVA is suitable for analyzing repeated measures data on the same population [30], while post hoc paired *t*-tests analyze specific differences between specific pairs of measures. Despite the small sample size of 23 participants, the data met the three assumptions required for these tests (independence, normality, and sphericity), confirming the appropriateness of the participant number for statistical analysis. In addition, Pearson correlation analysis was applied to determine the relationship between the measurement data and VAS ratings. These statistical methods were performed using the Pingouin library in Python 3.10.0 [31], and all analyses were performed at a significance level of *p* < 0.05.

## 3. Results

### 3.1. Experiment 1: Measurement of EDA for Each Pressure Condition

Table 1 shows the descriptive statistics of the EDA characteristics as a function of cuff pressure, with means and standard deviations. The mean SCL and SCR counts tended to increase in 10 out of 13 subjects and the maxSCR amplitude in 8 subjects. As shown in Table 2, RM ANOVA revealed that EDA parameters exhibited high F-values and low *p*-values as the pressure increased, suggesting a statistical significance of pressure on the EDA response. Table 3 shows the results of a paired *t*-test on EDA parameters, comparing different pressure levels. The statistical analysis indicated significant changes in the mean SCL across all comparisons, with *p*-values less than 0.05. For the max SCR amplitude, a significant difference was observed when comparing 10 kPa to 30 kPa, but not between 10 kPa and 20 kPa, indicating a differential response at higher pressures. SCR counts exhibited significant differences when the pressure increased from 10 kPa to 20 kPa and from 10 kPa to 30 kPa, but not between 20 kPa and 30 kPa.

Figure 3 shows the response of EDA parameters and the normalized VAS to increases in pressure. The mean SCL and the normalized VAS significantly increased with increasing pressure. The max SCR amplitude showed no significant increase between 10 and 20 kPa, and SCR counts did not significantly increase between 20 and 30 kPa, while others significantly increased.

The Pearson correlation coefficients between the normalized VAS and the EDA parameters indicated weak positive relationships for the mean SCL (r = 0.21), maximum SCR amplitude (r = 0.15), and SCR counts (r = 0.18).

### 3.2. Experiment 2: Measurement of StO_2_ and EDA for Different Pressures

Table 4 shows the mean and standard deviation of the bio-signal, categorized by pressure conditions. Among 10 subjects, the mean SCL increased in 5, the max SCR amplitude in 8, and SCR counts in 6. In Table 5, RM ANOVA revealed that EDA parameters and the decrease in StO_2_ exhibited high F-values and low *p*-values as the pressure increased, suggesting a statistical significance of pressure on EDA and StO_2_ responses. Table 6 shows the results of a paired *t*-test on EDA parameters and the decrease in StO_2_, comparing different pressure levels. EDA parameters exhibited statistically significant changes across all pressure conditions except between 20 and 30 kPa, while decreases in StO_2_ exhibited significant changes across all pressure levels.

Figure 4 shows the response of EDA parameters, the decrease in StO_2_, and the normalized VAS for pressure increases. All EDA parameters increased with increasing pressure, except for the group between 20 and 30 kPa, while the decrease in StO_2_ and the normalized VAS increased in all conditions.

Pearson correlations between the normalized VAS and all parameter values were as follows: r = 0.37 for the mean SCL, r = 0.032 for the maximum SCR amplitude, r = 0.11 for SCR counts, and r = 0.74 for the decrease in StO_2_. The decrease in StO_2_ showed a strong positive correlation, the mean SCL had a moderate positive correlation, and SCR counts had a weak positive correlation. However, no strong relationship existed between the max SCR amplitude and the normalized VAS.

## 4. Discussion

In this study, we aimed to quantify the comfort of the binding parts of rehabilitation robots in a sitting position by using bio-signals such as EDA and StO_2_ under different pressure conditions. This study employed a circular pneumatic cuff to apply continuous pressure to the binding parts, unlike previous studies that used digital algometers [32]. Three different pressure levels were applied to the thigh to measure both EDA and StO_2_. For EDA, the mean SCL, maximum SCR amplitude, and SCR counts were considered and correlated with the normalized VAS. The EDA characteristics tended to increase with increasing pressure, which is in agreement with previous studies [15,16], and this increase was found to be positively correlated with the normalized VAS. We found that, among the EDA characteristics, the mean SCL showed a relatively strong correlation and was statistically significant under the measured pressure conditions. Consistent with previous work, the decrease in StO_2_ obtained from the NIRS sensor was positively correlated with the normalized VAS [24]. While statistical significance was observed across all pressure conditions, the difference in values between 20 kPa and 30 kPa was less distinct than with other pressure changes. They suggested that beyond a threshold pressure of 20 kPa, the oxygen saturation in the tissue did not significantly decrease owing to vascular occlusion [24,33].

Some studies utilized machine learning to classify discomfort from EDA signals [8,9,17]. Based on these approaches, future research should focus on using machine learning to evaluate discomfort at binding parts. Beyond EDA, various bio-signals such as EMG (electromyography), HRV (heart rate variability), and RESP (respiration) were also applied in the study [34,35,36,37]. Even though it is quite challenging to quantify discomfort at the binding parts using HRV, HRV characteristic values exhibit a clear increasing trend with pressure increase. This observation aligns with a study that successfully performed binary classification of discomfort using HRV characteristics [38] and is further supported by research suggesting the potential of high-frequency HRV as a physiological marker in pain assessment [39]. Therefore, it would be important to apply more in-depth analysis methods to HRV in future studies.

In this study, changes in EDA were compared by applying different pressures for either 1 or 5 min to the binding parts of participants in a sitting posture. Comparing Table 2 with Table 5 shows that the F-values in Experiment 2 were smaller than those in Experiment 1. This reduction may reflect the homogenization of physiological responses over a prolonged stimulus period. It can be concluded that the initial variability in EDA responses, which could be amplified owing to individual differences in physiological and psychological conditions, tends to diminish over time because of adaptation to pressure. In addition, the number of subjects in Experiment 1 (n = 13) is larger than that in Experiment 2 (n = 10), which may affect the statistical results.

In practice, rehabilitation robots are likely to be worn for longer durations compared with the experimental conditions, and it is essential to consider the wearability factor when applying actual pressure in real-world settings. Given that a longer duration of pressure application may enhance the correlation of mean SCL, this parameter could potentially serve as a valuable indicator for assessing prolonged discomfort in users [37]. Furthermore, while our experiment involved stimulation of the right thigh only, it is important to acknowledge that actual wearable structures typically involve binding on both thighs. Since the binding parts in the actual rehabilitation robot are more complicated than the pressure cuff, it would be necessary to determine the binding parts under various conditions, such as in a standing state. Although this study did not use the actual binding parts of a rehabilitation robot, it represents a fundamental approach in which a pneumatic cuff-type binding is applied as a preliminary model.

The max amplitude of SCR and SCR counts did not show statistical significance. This may have resulted from the high dependence of the mean SCL on SCR since a high-pass filter was used in the decomposition of the EDA signal. We believe that the use of advanced EDA analysis tools, such as the cvxEDA algorithm, can significantly minimize this problem. The cvxEDA algorithm demonstrated a strong ability to detect SCRs from raw signals reliably [40], thereby enhancing the statistical significance of the maximum SCR amplitude and SCR counts. In addition, it is important to consider that not only noise or motion artifacts but also emotional stimuli can introduce changes in the EDA signal. In this study involving 23 healthy subjects, it is acknowledged that patients with lower-limb disabilities or disorders may present different bio-signal patterns [25,41]. Therefore, further research including these patients—who represent the actual users of rehabilitation robots—is essential for comprehensive analysis. Such efforts will not only validate and enrich our results but also contribute to the quantification of comfort in the binding parts of rehabilitation robots.

## 5. Conclusions

In the present study, the electrodermal activity (EDA) and tissue oxygen saturation (StO_2_) were measured to correlate with the discomfort measures due to the binding pressure in rehabilitation robots by means of a visual analog scale (VAS). The results showed that EDA parameters (mean SCL, max amplitude, and SCR counts) and the decrease in StO_2_ significantly increased with pressure increase. In addition, the use of the mean SCL and the decrease in StO_2_ showed a relatively strong correlation with the VAS, suggesting their appropriateness in quantifying comfort, as they demonstrated a clear link between increased binding pressure and significant physiological changes. Notably, a change in the EDA signal according to the pressure condition (10, 20, and 30 kPa) for 1 and 5 min was observed. A higher correlation coefficient was extracted from the mean SCL at pressure stimulation for 5 min compared with that for 1 min. Our findings showed that discomfort due to pressure stimulation could be measured by EDA and StO_2_ in comparison with the VAS, which would be useful for designing human–machine interfaces in rehabilitation robots.

## Figures and Tables

**Figure 1 sensors-24-00917-f001:**
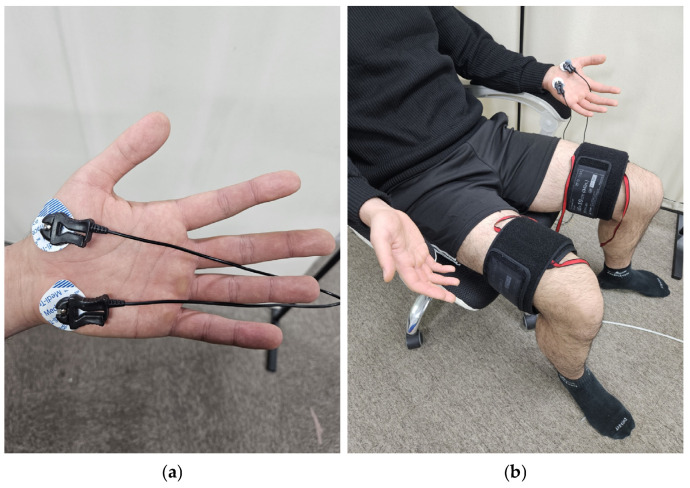
Equipment setting: (**a**) position of EDA sensor; (**b**) experimental setup.

**Figure 2 sensors-24-00917-f002:**
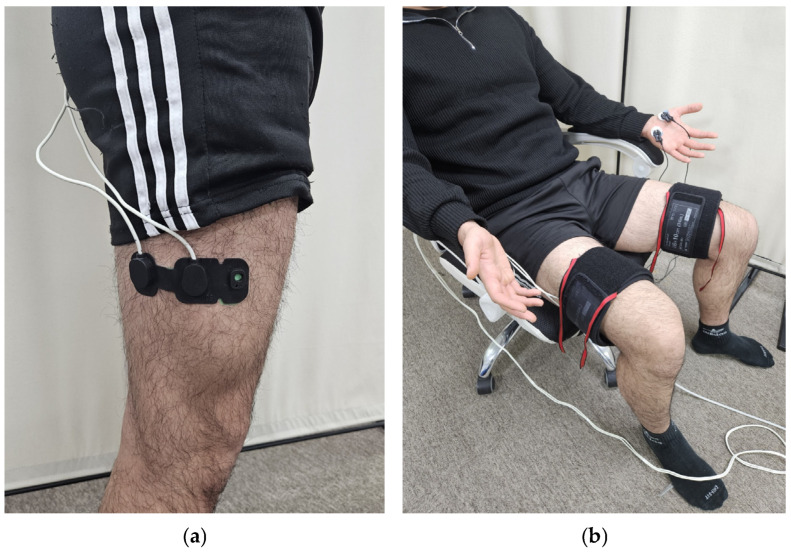
Equipment setting: (**a**) position of the NIRS sensor probe; (**b**) experimental setup.

**Figure 3 sensors-24-00917-f003:**
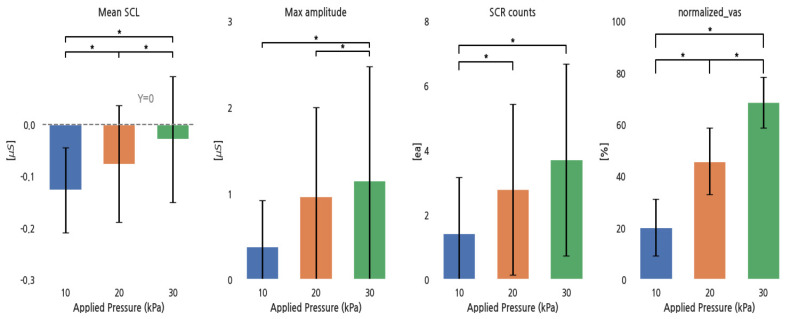
EDA parameters and normalized VAS for different pressure conditions. *: *p* < 0.05.

**Figure 4 sensors-24-00917-f004:**
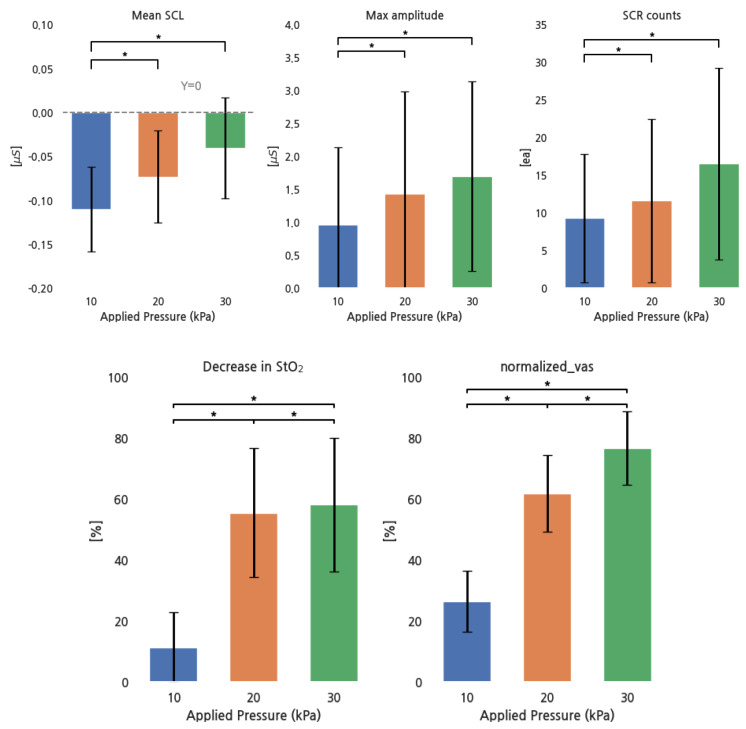
EDA parameters, StO_2_, and normalized VAS for different pressure conditions. *: *p* < 0.05.

**Table 1 sensors-24-00917-t001:** Mean ± SD of EDA parameters according to different pressure conditions.

Pressure [kPa]	Mean SCL [μS]	Max Amplitude [μS]	SCR Counts
10	−0.12 ± 0.23	1.08 ± 0.79	5.3 ± 3.8
20	−0.06 ± 0.35	1.40 ± 0.86	10.2 ± 7.9
30	−0.01 ± 0.67	2.09 ± 1.27	12.1 ± 1.4

**Table 2 sensors-24-00917-t002:** Statistical results for EDA parameters using RM ANOVA.

	Mean SCL	Max Amplitude	SCR Counts
F-value	16.684	8.455	10.790
*p*-value	0.000	0.002	0.001

**Table 3 sensors-24-00917-t003:** Statistical results for EDA parameters using a paired *t*-test.

Group	Mean SCL	Max Amplitude	SCR Counts
10 kPa vs. 20 kPa	0.008 *	0.086	0.011 *
10 kPa vs. 30 kPa	0.001 *	0.008 *	0.003 *
20 kPa vs. 30 kPa	0.002 *	0.005 *	0.051

*: *p* < 0.05.

**Table 4 sensors-24-00917-t004:** Mean ± SD of bio-signal parameters according to pressure conditions.

Pressure [kPa]	Mean SCL [μS]	Max Amplitude [μS]	SCR Counts	Decrease in StO_2_ [%]
10	−0.10 ± 0.04	1.47 ± 1.21	27.8 ± 22.5	11.46 ± 7.53
20	−0.07 ± 0.04	1.90 ± 1.53	34.7 ± 20.4	55.56 ± 20.53
30	−0.06 ± 0.05	2.27 ± 1.57	49.2 ± 35.2	58.19 ± 21.82

**Table 5 sensors-24-00917-t005:** Statistical results for bio-signal parameters using RM ANOVA.

	Mean SCL	Max Amplitude	SCR Counts	Decrease in StO_2_
F-value	6.574	8.264	8.281	55.292
*p*-value	0.007	0.003	0.003	0.000

**Table 6 sensors-24-00917-t006:** Statistical results for bio-signal parameters using a paired *t*-test.

Group	Mean SCL	Max Amplitude	SCR Counts	Decrease in StO_2_
10 kPa vs. 20 kPa	0.015 *	0.022 *	0.005 *	0.000 *
10 kPa vs. 30 kPa	0.016 *	0.010 *	0.006 *	0.000 *
20 kPa vs. 30 kPa	0.353	0.068	0.062	0.009 *

*: *p* < 0.05.

## Data Availability

The data presented in this study are available upon request from the corresponding author. The data are not publicly available because the authors are continuing the study.

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
