# Peer review of "Measurements of Electrodermal Activity, Tissue Oxygen Saturation, and Visual Analog Scale for Different Cuff Pressures"

_sensors, 2024, doi:10.3390/s24030917_

Round 1

Reviewer 1 Report

Comments and Suggestions for Authors

The aim of the study in this paper is to quantitatively investigate the comfort of a rehabilitation robot binding site by measuring EDA and StO2 in conjunction with VAS. The authors used EDA sensors and NIRS sensors for measurements, recorded multiple sets of data, and analyzed them statistically. Overall, the topic and experiments of this paper are innovative providing a reference for the design of human-robot interfaces for rehabilitation robots. However, in the course of the review, the following areas were noted as requiring further discussion and clarification:

(1)The aim of the authors' study was to analyze the effect of bonding force on the user by measuring the response of EDA, StO2 and VAS to different pressure intensities applied to the robotic bonding site, aiming at determining the optimal bonding force conditions. However, is there a lack of comparison between the three different pressures applied during the experiments when the sensors only selected the thigh area for measurement? The authors should consider analyzing other measurements, such as different parts, pressures, and actions, to make the experimental results more reliable.

(2)The authors used NIRS sensors to indirectly measure StO2. NIRS sensors are prone to limitations in certain application-specific scenarios, such as the depth of light penetration can vary depending on the subject's skin color, and the accuracy of the measurement results can be affected. The authors did not mention the relevant influencing factors during their analysis. Could the authors explore how to deal with the relevant confounding factors?

(3)Participants in the current study were selected from 23 healthy adults, including 21 men and 2 women. The study focuses on the comfort level of the rehabilitation robot's binding site, patients suffering from different types of diseases or injuries for the rehabilitation robot's binding site comfort and feedback is also different from healthy people. Should the author give the basis for selecting the testers and why those testers were selected?

(4)Whether the study, which focuses on quantifying the comfort of the binding site of the rehabilitation robot, should be labeled with a quantitative parameter in the conclusions section that gives the relationship between the increase in pressure and physiological changes, such as a certain trend exhibited by the physiological changes in the test subjects, so as to better support the research objectives.

Author Response

Dear Reviewer,

Thank you very much for taking the time to read our paper and providing detailed comments despite your busy schedule. We have addressed and provided responses to the points you raised, along with the corresponding revisions, in the attached document. Please have a look.

Thank you.

Reviewer 2 Report

Comments and Suggestions for Authors

The paper titled "Quantitative Analysis of Rehabilitation Robot Binding Parts using EDA and StO2" focuses on evaluating the comfort of binding parts in rehabilitation robots, and crucial human-machine interfaces. The study measures the physiological impacts of these parts on users, utilizing electrodermal activity (EDA) and tissue oxygen saturation (StO2). Two experiments were conducted with healthy subjects under different pressure conditions to assess discomfort levels. The study finds that EDA parameters and the decrease in StO2 significantly increase with pressure, suggesting that these measures can effectively quantify comfort in the binding parts of rehabilitation robots.

Here are some issues observed by the reviewer:

1.     Conducting a comprehensive power analysis to determine the appropriate sample size would significantly strengthen the study. It is essential to provide a clear rationale for the selection of 23 subjects, explaining how this number sufficiently meets the study's statistical needs and objectives.

2.     The study primarily involves healthy adults, which may not accurately represent the diverse range of users, including those with disabilities who typically use rehabilitation robots.

3.     The experiments were conducted over short durations, which might not capture long-term discomfort or physiological changes associated with the prolonged use of rehabilitation robots.

4.     The paper would be enhanced by the inclusion of a table detailing the demographics of the study participants. This table should encompass key demographic variables such as age, gender, and any other relevant characteristics.

5.     There is a typo on page, 7 line 233.

The paper presents a novel and practically relevant approach to assessing comfort in rehabilitation robots. Despite some limitations in terms of sample diversity and experiment duration, the study offers significant contributions to the field, particularly in providing a quantitative method for evaluating an aspect of robot design that directly impacts user experience.

Author Response

(The authors gave the same response as above.)

Round 2

Reviewer 1 Report

Comments and Suggestions for Authors

Dear author, I have read your reply.

It has responded to all the questions raised previously, clarified the corresponding experimental methodology and expression of conclusions, and I think it is acceptable for publication. However, in the conclusion section, is it possible to itemize.

Author Response

(The authors gave the same response as above.)
